# Prevalence and Clinical Characteristics of Visceral Involvement in HIV-Associated Kaposi Sarcoma: A Three-Year Retrospective Cohort Study at a Tertiary Care Center in Mexico

**DOI:** 10.3390/microorganisms13092187

**Published:** 2025-09-19

**Authors:** Emily Itzel Pecero-García, Juan Carlos Domínguez-Hermosillo, Yessica Sara Pérez-González, Juan Pablo Sánchez-Navarro, Mauricio Alfredo Ambriz-Alarcón, Natalia Jaime-Gómez, Sol Ramírez-Ochoa, Gabino Cervantes-Guevara, Berenice Vicente-Hernández, Francisco Javier Hernández-Mora, Enrique Cervantes-Pérez

**Affiliations:** 1Hospital de Infectología, Centro Médico Nacional “La Raza”, Mexico City 02990, Mexico; eipecerog92@gmail.com (E.I.P.-G.); cardomher@gmail.com (J.C.D.-H.); yessica_s_pg@hotmail.com (Y.S.P.-G.); jp_sanchezn@hotmail.com (J.P.S.-N.); 2División de Medicina, Hospital Civil de Guadalajara Fray Antonio Alcalde, Guadalajara 44280, Jalisco, Mexico; mau_ambriz@hotmail.com; 3Departamento de Medicina Interna, Hospital de Especialidades del Centro Médico Nacional de Occidente, Instituto Mexicano del Seguro Social, Guadalajara 44329, Jalisco, Mexico; jaimegomez.natalia@gmail.com; 4Departamento de Medicina Interna, Hospital Civil de Guadalajara Fray Antonio Alcalde, Guadalajara 44280, Jalisco, Mexico; sramirez@hcg.gob.mx (S.R.-O.); dra.berenicevicente@gmail.com (B.V.-H.); 5Departamento de Gastroenterología, Hospital Civil de Guadalajara Fray Antonio Alcalde, Centro Universitario de Ciencias de la Salud, Universidad de Guadalajara, Guadalajara 44280, Jalisco, Mexico; gabino_guevara@hotmail.com; 6Departamento de Bienestar y Desarrollo Sustentable, Centro Universitario del Norte, Universidad de Guadalajara, Colotlán 46200, Jalisco, Mexico; 7Departamento de Reproducción Humana, Bienestar y Desarrollo Infantil, Centro Universitario de Ciencias de la Salud, Universidad de Guadalajara, Guadalajara 44280, Jalisco, Mexico; frank.gine@gmail.com; 8Departamento de Obstetricia, Hospital Civil de Guadalajara Fray Antonio Alcalde, Guadalajara 44280, Jalisco, Mexico; 9Departamento de Clínicas, Centro Universitario de Tlajomulco, Universidad de Guadalajara, Tlajomulco de Zúñiga 45641, Jalisco, Mexico

**Keywords:** Kaposi sarcoma, HIV, AIDS, visceral Kaposi sarcoma

## Abstract

Despite advances in the understanding of Kaposi sarcoma (KS), research from resource-limited settings remains limited. This study aimed to estimate the proportion of epidemic visceral KS among Mexican people living with HIV (PLHIV) and to describe their clinical and biochemical characteristics. We included PLHIV with histopathologically confirmed KS who received care at the National Medical Center *La Raza* between March 2020 and February 2023. We calculated the prevalence of epidemic KS and epidemic visceral KS and analyzed clinical and biochemical variables potentially associated with visceral involvement. The prevalence of epidemic KS was 5.6%. Among these cases, 51.4% had visceral involvement, yielding an overall prevalence of 2.8%. Patients with epidemic visceral KS exhibited significantly higher rates of oral mucosal involvement and lower hemoglobin levels compared with those without visceral disease. These findings highlight the substantial burden of epidemic visceral KS in this population and should be confirmed in future studies with larger cohorts and robust designs aimed at identifying clinical and biochemical predictors of visceral involvement.

## 1. Introduction

Although early detection of human immunodeficiency virus (HIV) infection has improved internationally, late detection is still common, especially in countries with limited economic resources [1]. In Mexico, as of April 2022, 3636 people living with HIV (PLHIV) were detected, 482 of whom (13.256%) were in WHO stage 4 at the time of diagnosis. This late stage implies that the diagnosis of HIV infection was accompanied by the presentation of diseases defining acquired immunodeficiency syndrome (AIDS), including Kaposi’s sarcoma (KS), which has been reported in some cohorts of Mexican patients as the first or second most common neoplasm in PLHIV, representing up to 72% of the neoplasms presented in this population [2,3]. KS is defined as multifocal vascular neoplasia or hyperplasia of vascular or lymphatic endothelial origin caused by human herpes virus 8 (HHV8) infection that can affect the skin, viscera or other organs. It is characterized by capillary and perivascular connective tissue proliferation and has a variable course, ranging from minimal isolated mucosal or cutaneous involvement to systemic progression with multisystemic involvement [4]. Four clinical–epidemiological types of SK have been described: classic, endemic, epidemic, and iatrogenic (posttransplant) [5].

Until the HIV/AIDS epidemic, KS was considered a relatively rare disease worldwide, with a variable incidence across geographic regions, with its endemic forms in sub-Saharan Africa having the highest incidence rates. However, since the 1980s, following the identification of HIV infection and the consequent increase in the number of PLHIV, a different and more aggressive KS variant than the variants previously described in other populations began to be observed, as well as an association between the incidence of KS and HIV infection, especially in patients with AIDS in advanced stages [6,7]. This variant of KS is now known as epidemic KS or AIDS-related KS and is the second most common neoplasia, after non-Hodgkin lymphoma, in this patient population worldwide [5].

Since the advent of highly active antiretroviral therapy (HAART) in 1996, a reduction in the incidence of KS has been observed, which is more evident in high-income countries with access to treatment; for example, in the United States of America during the pre-HAART era (1991–1995), KS in patients with HIV was 500 to 2000 times more common than in the general population, particularly in those with high HIV viral loads and low CD4+ T-cell counts; after the advent of HAART, a consistent decrease of up to 6% of KS cases per year was observed [7,8]. An analysis of the Swiss cohort revealed that the incidence of KS was 33.3 per 1000 person-years between 1984 and 1986 and did not change significantly in subsequent periods until 1996 and 1998, when it fell to 5.1 per 1000 person-years and decreased to 1.4 per 1000 person-years in 1999–2001, remaining constant thereafter [9]. Beyond incidence and prevalence, with the optimization of antiretroviral treatment, the 5-year survival of KS patients also improved from 12.1% in 1980–1995 to 88% in the subsequent era [7,8,9].

Although the overall incidence of KS has decreased, this decrease occurred because of the lower incidence in Europe and North America [6]. In the global cancer statistics for 2022 reported by GLOBOCAN, 35,359 new cases and 15,911 new deaths from KS were identified worldwide, 45% of which were diagnosed in East African countries, where it was the second most common malignancy after prostate cancer. This region of the world also had the highest mortality rate during the same period [10]. A recent study conducted at a referral center in São Paulo, Brazil, with a Latin American population with a similar epidemiological profile to ours, reported a prevalence of epidemic KS of 6% [11].

Epidemic KS presents with a highly variable clinical course. It can present as an indolent disease limited to the skin with few lesions, but unlike other subtypes, up to 30–40% of patients have mucosal involvement, and 20–40% of patients have rapid spread to the gastrointestinal tract, lymph nodes, lungs and bones, which is characterized by an aggressive course and fatal outcome in many cases [12,13]. There are case reports of unusual locations of involvement for visceral forms, including the liver, musculoskeletal system, central and peripheral nervous system, larynx, eye, salivary glands, endocrine organs, heart, thoracic duct, urinary system and breast [14,15,16].

Since 1982, visceral involvement of KS has been associated with a poor prognosis, with a reduction in survival to only 1–2 years as opposed to the 10 to 25 years observed in KS with exclusively cutaneous involvement [17]. This worse prognosis continued even in the post-HAART era, although optimizing the immune status of patients has improved cutaneous disease symptoms. Visceral involvement requires additional therapeutic interventions, and there is a need for complementary studies in patients with suspected spread of KS to other organs, which represents a challenge given that visceral involvement is often asymptomatic or presents with nonspecific symptoms such as abdominal pain, dyspepsia, diarrhea or even gastrointestinal bleeding and intestinal obstruction [18]. In the case of pulmonary involvement, symptoms include cough, dyspnea, pleuritic pain, or hemoptysis, which are difficult to distinguish from other diseases caused by opportunistic pathogens [19].

Despite the increased knowledge of KS, there is still little information from countries with limited resources. In the case of Mexico, to our knowledge, there are no studies that describe the prevalence and characteristics of epidemic visceral KS in the Mexican population. The objective of this study was to determine the proportion of epidemic visceral KS in this population, as well as its clinical and biochemical characteristics.

## 2. Materials and Methods

To calculate the prevalence of KS and visceral KS, all patients over 18 years of age, regardless of sex, who sought care at the La Raza National Medical Center between March 2020 and February 2023 and who had a diagnosis of HIV infection confirmed by a positive result from at least two rapid diagnostic tests (RDTs) were included [20]. From this patient population, those diagnosed with KS, confirmed by histopathological examination, were included. Visceral KS was defined as KS with confirmed histopathological involvement of any visceral organ regardless of cutaneous manifestations [5]. The prevalence of KS and epidemic visceral KS in patients with HIV infection were calculated.

The prevalence was calculated as follows:$$Prevalence=\frac{\#\;of\;people\;in\;sample\;with\;characteristic\;(epidemic\;KS\;or\;visceral\;KS)}{Total\;\#\;of\;people\;in\;sample\;(people\;with\;HIV\;diagnosis)}$$

For the remainder of this study’s analyses, only patients diagnosed with HIV infection and KS treated at the La Raza National Medical Center between March 2020 and February 2023 were included. The inclusion criteria were as follows: patients of either sex over 18 years of age who sought medical care during the study period; a diagnosis of HIV infection confirmed by a positive result from at least two rapid diagnostic tests (RDTs) [20]; and a diagnosis of KS confirmed by histopathological results. The exclusion criterion was patients with a known diagnosis of KS prior to the study period.

The clinical information and laboratory results of the patients included in the study were collected from their digital medical records. The data collected for analysis included the following: sex, age, weeks since HIV diagnosis, clinical and laboratory variables associated with the management and follow-up of HIV infection, sites of involvement of the KS lesions, and results of relevant laboratory tests performed during hospitalization. Patients who were classified as having visceral KS were those diagnosed as such in their medical history, based on histopathological confirmation of visceral involvement; specifically, patients were not considered for the study if the diagnosis of visceral KS did not explicitly mention in the admission medical note or subsequent follow-up that the diagnosis had been confirmed by a biopsy of the visceral site with compatible histopathological results. While all records mentioning a diagnosis of visceral KS met these criteria, most lacked a complete description of the biopsy procedure or detailed histopathological results corresponding to visceral involvement.

Data from the clinical records were compiled into a database created in Microsoft Office Excel 2016 (Microsoft Corporation, Redmond, WA, USA). SPSS Statistics (version 29, IBM, Armonk, NY, USA) was used for data analysis.

For descriptive statistics of qualitative variables, absolute frequencies were calculated, and the data are presented as proportions and percentages. For quantitative variables, measures of central tendency and dispersion (median and interquartile range) were used for nonnormal distributions.

For the inferential statistics of the qualitative variables, the proportions of the categorical variables were placed in a contingency table to compare them using the Chi-square test. The odds ratio (OR) was also obtained with 95% confidence intervals (CIs) calculated using the Baptista–Pike method.

With respect to the inferential statistics of the quantitative variables, their normality within the Gaussian distribution was first assessed using the Kolmogorov–Smirnov test, and comparisons between groups were then made using the Mann–Whitney U test. A multivariate analysis was performed using binary logistic regression with a backward stepwise entry model with likelihood ratios and the Hosmer–Lemeshow goodness-of-fit test, which uses the clinical and biochemical variables that were significant in the bivariate analysis, as well as variables of relevance in other studies. The magnitude of the association between the variables and the development of visceral KS was determined by calculating ORs and their corresponding 95% CIs.

In all analyses, differences with *p* values less than 0.05 with a 95% CI were considered significant.

This study was conducted in accordance with the Declaration of Helsinki and Good Clinical Practice (GCP) guidelines. The study protocol was reviewed and approved by the Clinical Research and Bioethics Local Committee 3502 of the Centro Medico Nacional La Raza (approval code: R-2023-3502-021). Given the retrospective nature of the study, the requirement for informed consent was waived by the ethics committee.

## 3. Results

During the study period, 1314 patients with a confirmed diagnosis of HIV infection evaluated at La Raza were identified. Among these patients, 74 were diagnosed with KS, resulting in a prevalence of epidemic KS of 0.056 (5.6%). Among the 74 patients with epidemic KS, 38 (51.4%) were diagnosed with visceral KS, resulting in a prevalence of epidemic visceral KS of 0.028 (2.8%).

For analysis, the patients were divided into two groups: patients with nonvisceral epidemic KS (n = 36, 48.6%) and patients with visceral epidemic KS (n = 38, 51.4%). Patients with epidemic visceral KS had greater oral mucosal involvement and lower serum hemoglobin levels, both of which were statistically significant (*p* = 0.01 and *p* = 0.02, respectively). Table 1 shows the characteristics and clinical history of the patients included in the study, as well as the results of laboratory tests taken upon hospital admission.

Among the patients with epidemic visceral KS (n = 38), 14 (36.8%) had lung involvement, 27 (71%) had upper gastrointestinal involvement, 9 (23.7%) had colon involvement, 13 (34.2%) had lymph node involvement, and 3 (7.9%) had involvement at sites other than those mentioned above. Figure 1 shows a graphical representation of these results.

Coinfections in addition to KS were identified in 44 (59%) of the patients included in the study, the most common being syphilis, followed by cytomegalovirus (CMV) infection; 19 (25.7%) patients had more than one coinfection. Table 2 shows the coinfections present in the study patients.

Table 3 shows the results of the multivariate binary regression analysis, which included the variables found to be significant in the previous analyses (hemoglobin and mucosal involvement), as well as some clinically relevant variables in KS patients.

## 4. Discussion

The primary objective of our study was to evaluate the prevalence of epidemic KS and epidemic visceral KS in our patient population, as well as to evaluate the proportion of visceral KS in patients with HIV infection and KS. In this cohort, we reported a prevalence of epidemic KS of 5.8%, which is very similar to that reported by Tancredi et al. [11] in a study conducted in Brazil, where they reported a prevalence of 6%; this prevalence is slightly higher than that reported by Semango et al. [21] in a study in Tanzania, which was 4.5%. Although this prevalence is much higher than that reported in countries such as the United States, which reported a prevalence of 0.20% for epidemic KS in 2015 [8], it coincides with the global increase in KS cases observed in the GLOBOCAN 2022 results, which reported a prevalence of 9.5% for KS (in all its clinical variants) for Latin America and the Caribbean [10]. The discrepancy in the prevalence of epidemic KS in resource-limited countries may be related to the lack of greater public health campaigns for the early detection of HIV infection [22].

Regarding visceral epidemic KS, we found a prevalence of 2.8%, and the proportion of patients having visceral involvement was 51.4% (38 of 74 patients) among all patients diagnosed with epidemic KS. Although visceral involvement in KS is considered rare, it occurs more frequently in epidemic KS; however, the reported frequencies are highly variable. In the study by Bower et al. [23], a prospective study conducted in London, visceral involvement was reported in 15% of patients with epidemic KS; this is much lower than the 62.7% of epidemic KS patients with visceral involvement reported by Pires et al. [19] in a retrospective study conducted in Brazil. This result is similar to that reported by Parente et al. [24] in a study conducted in Italy, which reported that the proportion of epidemic KS patients with visceral involvement was 51%. These differences could be due to the lack of clinical symptoms associated with visceral involvement, which, even when it occurs, is nonspecific and easily confused with the clinical manifestations of other diseases associated with HIV or AIDS; however, it seems that visceral involvement is much more frequent than is considered, and 33–70% of patients with cutaneous KS have visceral involvement at the time of autopsy [25].

In this study cohort, the most frequent site of visceral involvement was the upper gastrointestinal tract (71%), followed by the pulmonary tract (50%). This finding is similar to that reported by Parente et al. [24], who reported that 51% of their patients with epidemic visceral KS had upper gastrointestinal tract involvement. Pires et al. [19] reported that the most frequent site of visceral involvement (75% of patients with epidemic visceral KS) was the gastric system; however, they reported that only 9.4% of their patient population had pulmonary involvement. The high frequency of pulmonary involvement in our patients could be because the study by Pires et al. [19] included a small number of patients (15 patients in total); another explanation may be related to the type of diagnostic approach for patients with HIV infection and KS that is performed in each hospital, which is supported by the finding that visceral involvement is generally asymptomatic and probably underdiagnosed [25]. Even so, studies such as that of Rezende et al. [26] report a proportion of up to 55% of epidemic visceral KS patients with pulmonary involvement and gastrointestinal involvement.

Regarding clinical and laboratory characteristics, we found that patients with visceral involvement appeared to have less cutaneous involvement (9.8 times lower) than patients with epidemic nonvisceral KS did (97% vs. 78%, *p* = 0.02; epidemic nonvisceral KS vs. epidemic visceral KS, respectively); however, the broad CI highlights the need to confirm these results with studies with a larger number of patients.; although visceral KS without cutaneous manifestations is uncommon [25], we believe this highlights the importance of reevaluating the need to intentionally search for visceral KS in patients with newly diagnosed AIDS, even those with nonspecific or absent symptoms.

Patients with epidemic visceral KS in our study also had a greater frequency (0.68 times greater) of oral mucosal involvement than patients without visceral involvement in both bivariate (*p* = 0.01) and multivariate analyses (*p* = 0.04); this has been reported as a common site of involvement in patients with visceral KS [27]. The laboratory results revealed no significant differences in CD4+ T lymphocytes, viral load or platelets; however, we detected significant differences in serum hemoglobin values in both the bivariate analysis (11.8 vs. 10.2 g/dL, *p* = 0.02, nonvisceral epidemic KS vs. visceral epidemic KS, respectively) and the multivariate analysis (*p* < 0.01), possibly because of non-specific clinical manifestations of visceral KS, such as gastrointestinal bleeding [25]. In the study by Polizzotto et al. [28], low hemoglobin levels (<13.7 g/dL) were associated with increased mortality in patients with epidemic KS. In our population, both patients with nonvisceral KS and those with visceral KS presented low hemoglobin levels; however, patients with visceral KS presented significantly lower levels. This could lead to increased mortality in these patients; however, this finding should be verified in future studies. We did not find any significant differences in the proportion of patients receiving antiretroviral therapy (ART) between groups. However, this variable was only considered a baseline characteristic of the population. This finding suggests that the observed differences between groups are independent of ART use. Nevertheless, due to the retrospective design of the study, we cannot draw conclusions about the possible effects of ART on visceral Kaposi sarcoma evolution or presentation.

Grabar et al. and the aforementioned work by Rezende et al. revealed that a CD4+ T lymphocyte count <200 cells/mm^3^ is a risk factor for epidemic KS [26,29]. In addition, Grabar et al. stratified patients according to their CD4+ T lymphocyte count into ≤50, 51–100 and 101–200 cells/mm^3^ groups and reported that, overall, the highest risk for KS was in patients with less than 50 cells/mm^3^, who had a 5-fold greater risk, and these patients had a 12-fold greater risk of visceral KS [29]. In our population, we found no statistically significant difference between the median CD4+ T lymphocyte count and the presence or absence of visceral KS, but this may be due to the sample size of our study and should be confirmed in future prospective studies.

Regarding coinfections, in our patient population, we found no significant differences between patients with nonvisceral epidemic KS and those with visceral epidemic KS. This may be related to the fact that both groups had similar levels of CD4+ T lymphocytes.

The high incidence of epidemic visceral KS in our patient population makes it relevant to evaluate the need for a deliberate search for visceral involvement in all patients with epidemic KS. However, due to the limitations of working in a setting with limited healthcare resources, the necessary diagnostic procedures are not always feasible [30]. Nevertheless, our results support the importance of conducting studies that evaluate clinical outcomes and risks associated with the clinical presentation of epidemic visceral KS in our population. These studies would help us optimize the use of resources for diagnostic procedures, such as endoscopy, in patients who may benefit most from a definitive KS diagnosis, which is especially relevant in high-incidence populations such as ours.

Our study has several limitations, which are important to mention. Due to the retrospective study design, we were limited by the information available in patients’ medical records and we did not perform a sample size calculation, which would have given us greater certainty regarding the statistical power of the results presented. Therefore, any differences between groups do not necessarily indicate a causal relationship and must be confirmed in subsequent studies. Furthermore, we cannot guarantee that all patients were studied with the same laboratory test profile or that biopsies were obtained uniformly from sites of possible visceral involvement, which could have been influenced by disease severity; therefore, since there is no histopathological description and method used for biopsy collection in the majority of medical records of patients diagnosed with visceral involvement in our study, the prevalence of visceral KS in our patient population may be under- or overestimated, this is because the diagnostic confirmation information was based solely on the description in the admission or follow-up medical note. Importantly, although early detection of HIV has been increasing in Mexico, our national limitation in health care resources makes it likely that KS and HIV are underdiagnosed [1]. In addition, patients who seek medical care generally arrive at an advanced stage of HIV infection or AIDS, which may introduce bias regarding the study population that was included. It is also important to note that the majority of our study population is male (97% of the total sample and 100% of the group presenting with epidemic visceral KS). Therefore, our results may not reflect the clinical characteristics of epidemic KS in women. To our knowledge, most studies conducted in non-African populations have a high preponderance of men [8,19,23,24], and only reports of clinical differences in the presentation of epidemic KS exist in the African population. However, due to the clinical and virological differences of the HIV epidemic in this population, the results may not be entirely comparable [31]. Despite the limitations described above, our results highlight the importance of screening for KS in PLHIV, which has a high prevalence in our population, even in its visceral form. Furthermore, to our knowledge, this is the first study to analyze the proportion of epidemic visceral KS in Mexican patients.

## 5. Conclusions

The prevalence of epidemic (HIV-related) KS in our patient population was 5.6%. Among these patients, 51.4% had visceral KS involvement, resulting in an overall prevalence of 2.8%. Patients with visceral epidemic KS had significantly greater oral mucosal involvement and lower serum hemoglobin levels than patients with nonvisceral epidemic KS did. Our results should be confirmed in future studies with larger sample sizes and a design aimed at identifying possible variables associated with visceral epidemic KS; however, these results highlight the importance of performing diagnostic studies to detect visceral involvement in patients with epidemic KS.

## Figures and Tables

**Figure 1 microorganisms-13-02187-f001:**
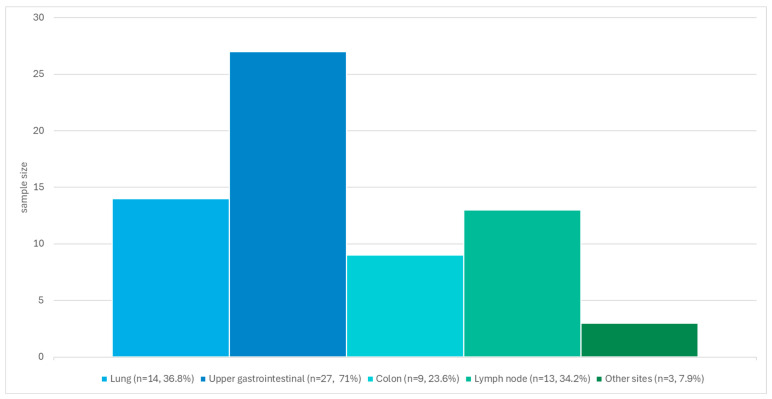
Anatomical sites of visceral involvement in patients with visceral epidemic Kaposi sarcoma (*n* = 38).

**Table 1 microorganisms-13-02187-t001:** Clinical and laboratory characteristics of the patients included in the study.

		Total Sample (*n* = 74)	Nonvisceral Epidemic KS (*n*= 36)	Visceral Epidemic KS (*n*= 38)	Odds Ratio (CI 95%)	*p*
		No. (%)	No. (%)	No. (%)		
Sex	Male	72 (97.3)	34 (94.4)	38 (100)	-	0.23
	Female	2 (2.7)	2 (5.6)	0		
Initiation of ART prior to admission	Yes	21 (28.4)	8 (22.2)	13 (34.2)	0.54 (0.18–1.60)	0.30
	No	53 (71.6)	28 (77.8)	25 (65.8)		
Skin involvement	Yes	65 (87.8)	35 (97.2)	30 (78.9)	9.33 (1.22–106.1)	0.02
	No	9 (12.2)	1 (2.8)	8 (21.1)		
Oral mucosa involvement	Yes	41 (55.4)	15 (41.7)	26 (68.4)	0.32 (0.13–0.82)	0.01
	No	33 (44.6)	21 (58.3)	12 (31.6)		
ART at admission	BIC/TAF/FTC	17 (23)	8 (22.2)	9 (23.7)	0.92 (0.31–2.77)	0.88
	DTG/ABC/3TC	2 (2.7)	0	2 (5.3)	-	0.19
	EFV/TDF/FTC	2 (2.7)	0	2 (5.3)	-	0.19
Patients classified according to their HIV viral load	Undetectable (<40 copies/mm^3^)	12 (16.3)	6 (16.7)	6 (15.8)	1.06 (0.28–3.88)	0.19
	40–200 copies/mm^3^	3 (4.1)	1 (2.8)	2 (5.3)	0.51 (0.03–4.61)	0.73
	201–100,000 copies/mm^3^	27 (36.5)	10 (27.8)	17 (44.7)	0.47 (0.19–1.22)	0.23
	>100,000 copies/mm^3^	32 (43.2)	19 (52.8)	13 (34.2)	2.14 (0.88–5.61)	0.34
		Median (IQR)	Median (IQR)	Median (IQR)		
Age in years		31 (26–36)	31 (27–41)	31 (26–34)	-	0.34
Weeks since diagnosis of HIV infection		4 (1.7–26)	3.5 (1–32)	7.5 (2–19.2)	-	0.46
Hemoglobin (g/dL)		10.9 (9.6–12.6)	11.8 (10.0–13.5)	10.2 (8.4–12.1)	-	0.02
Platelets (×10^9^/L)		205 (107–281)	223 (161–281)	180 (72–284)	-	0.20
TCD4+ lymphocytes (cel/µL)		48 (17–93)	37 (16–176)	69 (25–127)	-	0.14
HIV Viral load (copies/mm^3^)		76,399 (1010.2–284,585)	127,225 (4863–373,421)	52,935 (1010.2–266,241.5)	-	0.23
HIV viral load (log)		4.84 (2.99–5.45)	5.0 (3.0–5.57)	4.72 (2.99–5.42)	-	0.29

KS = Kaposi sarcoma, IQR = Interquartile range, HIV = Human immunodeficiency virus, ART = Antiretroviral therapy, BIC = bictegravir, TAF = tenofovir alefenamide, FTC = emtricitabine, DTG = Dolutegravir, ABC = Abacavir, 3TC = Lamivudine, EFV = Efavirenz, TDF = Tenofovir disoproxil fumarate.

**Table 2 microorganisms-13-02187-t002:** Coinfections present in the patients with epidemic KS included in the study.

	Total Sample (*n* = 74)	Nonvisceral Epidemic KS (*n* = 36)	Visceral Epidemic KS (*n* = 38)	Odds Ratio (CI 95%)	*p*
	*n* (%)	*n* (%)	*n* (%)		
Patients with one or more coinfections	44 (59.5)	25 (69.4)	19 (50)	2.27 (0.91–5.63)	0.08
CMV infection	14 (18.9)	8 (22.2)	6 (16.8.)	1.52 (0.51–5.29)	0.48
Non-Tb mycobacterium infection	4 (5.4)	2 (5.6)	2 (5.3)	1.05 (0.15–7.04)	0.67
Tb	6 (8.1)	3 (8.3)	3 (7.9)	1.06 (0.22–5.01)	0.63
Histoplasmosis	2 (2.7)	1 (2.8)	1 (2.6)	1.05 (0.10–10.63)	0.74
Cryptococcosis	3 (4.1)	0	3 (7.9)	-	0.13
Syphilis	18 (24.3)	10 (27.8)	8 (21.1)	1.44 (0.50–4.29)	0.50
HCVAb +	4 (5.4)	2 (5.6)	2 (5.3)	1.05 (0.15–7.04)	0.95
HBsAg +	5 (6.7)	2 (5.5)	3 (7.9)	0.68 (0.11–3.54)	0.68
HBcAg -	11 (14.9)	6 (16.6)	5 (13.1)	1.32 (0.37–4.38)	0.67

KS = Kaposi sarcoma, CMV = Cytomegalovirus, Tb = Tuberculosis, HCV Ab = Hepatitis C antibody, HBsAg = Hepatitis B surface antigen, HBcAG = Hepatitis B core antigen.

**Table 3 microorganisms-13-02187-t003:** Results of the multivariate analysis for variables associated with epidemic visceral KS.

	OR (CI 95%)	*p*
Oral mucosa involvement	3.05 (1.01–9.2)	0.04
Hemoglobin	1.4 (1.09–1.8)	<0.01
TCD4+ lymphocytes	0.9 (0.9–1.0)	0.1
Prescence of coinfections	0.3 (0.1–1.1)	0.09

OR = odds ratio, CI = confidence interval.

## Data Availability

The data presented in this study are not publicly available due to privacy reasons. Requests to access the data should be directed to the corresponding author.

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
