# Peer review of "Prevalence and Clinical Characteristics of Visceral Involvement in HIV-Associated Kaposi Sarcoma: A Three-Year Retrospective Cohort Study at a Tertiary Care Center in Mexico"

_microorganisms, 2025, doi:10.3390/microorganisms13092187_

Round 1
Reviewer 1 Report
Comments and Suggestions for Authors
In general, this is an interesting study which aimed to estimate the proportion of epidemic visceral KS among Mexican people living with HIV (PLHIV) and to describe their clinical and biochemical characteristics. They found that the prevalence of epidemic KS was 5.6%. Among these cases, 51.4% had visceral involvement, yielding an overall prevalence of 2.8%. Patients with epidemic visceral KS exhibited significantly higher rates of oral mucosal involvement and lower hemoglobin levels compared with 40 those without visceral disease. Such clinical studies from resource-limited settings will help to increasing our understanding of KS in the era of HAART. I have a few questions about this manuscript:
1) Page 2, Line 50-51, “3,636 people living with HIV (PLHIV) were detected, 482 of whom (13.25%) were in WHO stage 4 at the time of diagnosis”, based on the percentage, I think the total number of PLHIV here is wrong, please correct it.
2) Page 2, Line 75, “SK in HIV-infected patients was 500 to 2000 times more common”, “SK” should be “KS”.
3) Page 4, Line 177-179, “a prevalence of epidemic KS of 0.56 (5.6%); epidemic visceral KS of 0.28 (2.8%)”. The numbers should be 0.056 and 0.028? Please check it.
4) From Table 1, I saw most of patients involved in this study is male (97%). Do they think this sex difference may affect their results in this study? What about comparing with other published studies they have discussed (sex ratio)?
5) Are all of these patients under HAART or not? Need some explanation or discussion.
Author Response
1) Page 2, Line 50-51, “3,636 people living with HIV (PLHIV) were detected, 482 of whom (13.25%) were in WHO stage 4 at the time of diagnosis”, based on the percentage, I think the total number of PLHIV here is wrong, please correct it.
Response: We reviewed the information regarding the number of PLHIV and the percentage of patients in stage 4. Apparently, the reference we cited rounded the percentage to two decimal places (13.25%), but the accurate percentage is 13.256%. We have corrected the text accordingly.
2) Page 2, Line 75, “SK in HIV-infected patients was 500 to 2000 times more common”, “SK” should be “KS”.
Response: We appreciate your comment. We have corrected the text accordingly..
3) Page 4, Line 177-179, “a prevalence of epidemic KS of 0.56 (5.6%); epidemic visceral KS of 0.28 (2.8%)”. The numbers should be 0.056 and 0.028? Please check it.
Response: We sincerely thank you for this valuable observation. You are correct, the omission of a zero after the decimal point was an oversight on our part. The correct values are 0.056 (5.6%) and 0.028 (2.8%), and the text has been revised accordingly.
4) From Table 1, I saw most of patients involved in this study is male (97%). Do they think this sex difference may affect their results in this study? What about comparing with other published studies they have discussed (sex ratio)?
Response: Although women are more likely to be PLHIV globally, this is primarily due to the high prevalence in certain regions of Africa. In the remaining populations, however, there is a marked difference in prevalence, with men being more affected. From our investigation, we found no studies similar to ours suggesting a difference in KS clinical presentation related to sex. Most studies include a disproportionate number of men. In Mexico, men who have sex with men have the highest HIV prevalence; therefore, we expected that most of our study participants would be men (there are also no reports that sex affects KS incidence). We found a study conducted in Uganda (Phipps et al., 2010) that mentions marked differences in the clinical presentation of epidemic KS were observed between women and men in this region. However, our results are not entirely comparable due to the significant clinical and virological differences in this population's epidemic. We have added information about this in the discussion.
5) Are all of these patients under HAART or not? Need some explanation or discussion.
Response: As can be seen in Table 1, due to the retrospective design of the study, we only considered whether patients were receiving ART at the time they sought care at our hospital. This was done solely to rule out the possibility that differences in the clinical presentation of KS in our population were due to immunosuppression and lack of treatment (these were baseline conditions). As mentioned in the discussion, there were no significant differences in CD4 counts, viral load, or the proportion of patients receiving therapy at the time of seeking care in our study. Following your suggestion, we have added a clarification to the discussion.
Reviewer 2 Report
Comments and Suggestions for Authors
This retrospective cohort study examined the prevalence and clinical characteristics of visceral Kaposi sarcoma (KS) among people living with HIV (PLHIV) treated at the National Medical Center “La Raza” in Mexico between 2020–2023. Out of 1,314 PLHIV, 74 (5.6%) were diagnosed with epidemic KS, and 38 (51.4%) of them had visceral involvement (2.8% overall prevalence). Visceral KS patients more frequently showed oral mucosal involvement and had significantly lower hemoglobin levels than those without visceral disease. Common sites of visceral involvement included the gastrointestinal tract (71%) and lungs (50%). Coinfections (e.g., syphilis, CMV) were common but not significantly different between groups. Epidemic visceral KS represents a considerable burden in this Mexican cohort. Oral mucosal involvement and anemia may serve as clinical indicators of visceral disease. The authors recommend larger, prospective studies to identify predictors and improve early detection.
Suggestions for Review
-The inclusion/exclusion criteria are clear, but the description of biopsy and diagnostic methods for visceral involvement could be expanded to address possible variability. Please provide more detail on how missing or inconsistent medical records were handled.
As regarding statistical analysis, the multivariate model is informative, but sample size limitations should be discussed more in relation to statistical power. Consider including effect size interpretations for clinical relevance (not just p-values).
In results presentation:
-Tables are clear, but summarizing the most relevant findings (hemoglobin levels, oral mucosa involvement) in the text would help highlight key differences.
-Figures or charts (e.g., distribution of visceral involvement sites) could improve readability.
In Discussion:
-The discussion contextualizes findings well with international literature but could expand on implications for clinical practice in resource-limited settings.
-Consider discussing whether systematic screening (e.g., endoscopy) is feasible or recommended in similar populations.
For limitations:
I think limitations are acknowledged, but more emphasis on diagnostic variability and underdiagnosis bias would strengthen transparency.
Author Response
- The inclusion/exclusion criteria are clear, but the description of biopsy and diagnostic methods for visceral involvement could be expanded to address possible variability. Please provide more detail on how missing or inconsistent medical records were handled.
- Response: As you suggested, we added a detailed description of the medical record review process and how we determined which patients with visceral KS were included in the study. We added the limitations to emphasize this point.
- As regarding statistical analysis, the multivariate model is informative, but sample size limitations should be discussed more in relation to statistical power. Consider including effect size interpretations for clinical relevance (not just p-values).
- Response: Following your recommendation, we expanded the limitations section to emphasize the small sample size and lack of formal sample size calculation. We have also added effect sizes for qualitative variables to the results tables and added text referring to the effect size results to the discussion section. We have modified the description of the statistical plan in the Materials and Methods section to reflect these changes. We have modified the structure of Table 1 to improve the clarity and presentation of the data.
- Tables are clear, but summarizing the most relevant findings (hemoglobin levels, oral mucosa involvement) in the text would help highlight key differences.
- Response: As you suggested, we added the relevant results from Table 1 to the text to highlight them more effectively.
- Figures or charts (e.g., distribution of visceral involvement sites) could improve readability.
- Response: As you recommended, we added a figure showing the results of the anatomical distribution of visceral involvement.
- The discussion contextualizes findings well with international literature but could expand on implications for clinical practice in resource-limited settings.
- Response: As you recommended, we added a brief text about clinical practice in resource-limited settings to the discussion section.
- Consider discussing whether systematic screening (e.g., endoscopy) is feasible or recommended in similar populations.
- Response: We have added a brief text to the discussion regarding clinical practice in resource-limited settings. In it, we mention the difficulties of systematic screening and diagnostic procedures.
- I think limitations are acknowledged, but more emphasis on diagnostic variability and underdiagnosis bias would strengthen transparency.
- Response: We have added clarifications to the discussion to improve transparency regarding the limitations and facilitate interpretation of the results.
Round 2
Reviewer 1 Report
Comments and Suggestions for Authors
The authors have correctly addressed all of my comments